METHODS AND PROTOCOLS
Applied and Environmental Science
# Visualizing Microbial Community Dynamics via a Controllable Soil Environment

Arunima Bhattacharjee,[a] Dusan Velickovic,[a] Thomas W. Wietsma,[a] Sheryl L. Bell,[a] Janet K. Jansson,[b] Kirsten S. Hofmockel,[a,c]
Christopher R. Anderton[a]

[a]Environmental Molecular Sciences Division, Pacific Northwest National Laboratory, Richland, Washington, USA
[b]Biological Sciences Division, Earth and Biological Sciences Directorate, Pacific Northwest National Laboratory, Richland, Washington, USA
[c]Department of Ecology, Evolution and Organismal Biology, Iowa State University, Ames, Iowa, USA

**ABSTRACT** Understanding the basic biology that underpins soil microbiome interactions is required to predict the metaphenomic response to environmental shifts. A significant knowledge gap remains in how such changes affect microbial community dynamics and their metabolic landscape at microbially relevant spatial scales. Using a custom-built SoilBox system, here we demonstrated changes in microbial community growth and composition in different soil environments (14%, 24%, and 34% soil moisture), contingent upon access to reservoirs of nutrient sources. The SoilBox emulates the probing depth of a common soil core and enables determination of both the spatial organization of the microbial communities and their metabolites, as shown by confocal microscopy in combination with mass spectrometry imaging (MSI). Using chitin as a nutrient source, we used the SoilBox system to observe increased adhesion of microbial biomass on chitin islands resulting in degradation of chitin into *N*-acetylglucosamine (NAG) and chitobiose. With matrix-assisted laser desorption/ionization (MALDI)-MSI, we also observed several phospholipid families that are functional biomarkers for microbial growth on the chitin islands. Fungal hyphal networks bridging different chitin islands over distances of 27 mm were observed only in the 14% soil moisture regime, indicating that such bridges may act as nutrient highways under drought conditions. In total, these results illustrate a system that can provide unprecedented spatial information about interactions within soil microbial communities as a function of changing environments. We anticipate that this platform will be invaluable in spatially probing specific intra- and interkingdom functional relationships of microbiomes within soil.

**IMPORTANCE** Microbial communities are key components of the soil ecosystem. Recent advances in metagenomics and other omics capabilities have expanded our ability to characterize the composition and function of the soil microbiome. However, characterizing the spatial metabolic and morphological diversity of microbial communities remains a challenge due to the dynamic and complex nature of soil microenvironments. The SoilBox system, demonstrated in this work, simulates an ~12-cm soil depth, similar to a typical soil core, and provides a platform that facilitates imaging the molecular and topographical landscape of soil microbial communities as a function of environmental gradients. Moreover, the nondestructive harvesting of soil microbial communities for the imaging experiments can enable simultaneous multiomics analysis throughout the depth of the SoilBox. Our results show that by correlating molecular and optical imaging data obtained using the SoilBox platform, deeper insights into the nature of specific soil microbial interactions can be achieved.

**KEYWORDS** SoilBox, metaphenome, mass spectrometry imaging, chitin, soil moisture, fungal bridging, soil ecosystem, community metabolism, drought, hyphae, imaging, microorganism

This article followed an open peer review process. The review history can be read here.

Address correspondence to Christopher R. Anderton, Christopher.Anderton@pnnl.gov.

Soil microbial communities are drivers of essential processes that are crucial in maintaining the health of the soil ecosystem, where they perform key functions in regulating the biogeochemical cycling of carbon and micronutrients (1). Consequently, considerable research has been invested in studying the composition and potential functions of the soil microbiome under different environmental conditions of moisture, temperature, nutrient availability, etc., that relate to global ecosystem transformations (2). Understanding how soil microbiomes respond to environmental perturbations has become increasingly realized through harnessing multiple omics technologies such as metagenomics, metatranscriptomics, and metaproteomics (1). However, these sampling techniques are destructive and often ignore the influence of microscale soil heterogeneity on microbial community processes. It is known that community morphology and function at the microscale resolution are greatly influenced by soil pH (3), bioavailable organic content (4), temperature (5), and redox state (6). Adding to the complexity of the microscale soil environment, the size and distribution of soil aggregates heavily regulate many of the abovementioned soil attributes (7, 8). These issues combined make it extremely difficult to identify direct structure-function relationships of soil microbial communities using conventional omics techniques that measure bulk processes. Consequently, knowledge gaps exist between the roles of different microbial species and the biochemical networks in which they participate that influence large-scale changes in the soil ecosystem in response to various environmental conditions.

To address the challenges of studying the soil microbiome in its native state and at microscale resolution, we built a SoilBox system that simulates an ~12-cm-deep soil core with individually addressable ports (i.e., windows) that permit the ability to spatially and temporally resolve soil microbial processes in living soil. The SoilBox represents a soil ecosystem, where microbial communities from native soil can be probed within a regulated environment. We demonstrate the value of this system for viewing soil microbial community development and metabolism as a function of different environmental conditions (i.e., moisture regimes) that can be manipulated in a laboratory setting.

Previously, the spatiotemporal dynamics of microbial community development and growth at microscale resolution have been studied using confocal microscopy (9–12). In addition, advanced chemical imaging techniques, like mass spectrometry imaging (MSI) methods, have been employed to examine complex molecular architectures of mammalian tissues (13, 14) and microbial communities on agar surfaces (15–18). Inspired by such examples, we characterized the community structure and metabolome of soil ecosystems in the SoilBox, using confocal microscopy and matrix-assisted laser desorption/ionization (MALDI)-MSI. These correlated imaging platforms provided us with the potential to link spatial metabolomics data with spatial taxonomical information from diverse soil microbial communities that were exposed to different environmental conditions, such as changes in moisture, temperature, and nutrient sources.

## RESULTS AND DISCUSSION

The SoilBox was designed to characterize changes in diversity and composition of microbial communities with increases in depth from the soil-air interface. The soil depth was maintained to 11.6 cm within each SoilBox (Fig. 1a), which is consistent with the surface soil depth that normally maintains the highest microbial biomass and metabolic activity (19). To emulate the patchy bioavailability of carbon sources typical of soil ecosystems, chitin islands were introduced at reproducible distance periodicities on the indium tin oxide (ITO)-coated slides (Fig. 1c; see also Fig. S1 in the supplemental material) and placed in contact with the soil. Chitin was chosen as our representative carbon source for development and validation of this method due to (i) its global abundance in soil ecosystems, where microbial degradation of chitin contributes toward carbon (and nitrogen) cycling in the environment (20, 21), and (ii) known methods for chitin thin film fabrication on glass surfaces (22).

We explored how moisture availability regulates microbial physiology, access to resources, and metabolic interactions within the simulated soil environment. We tested

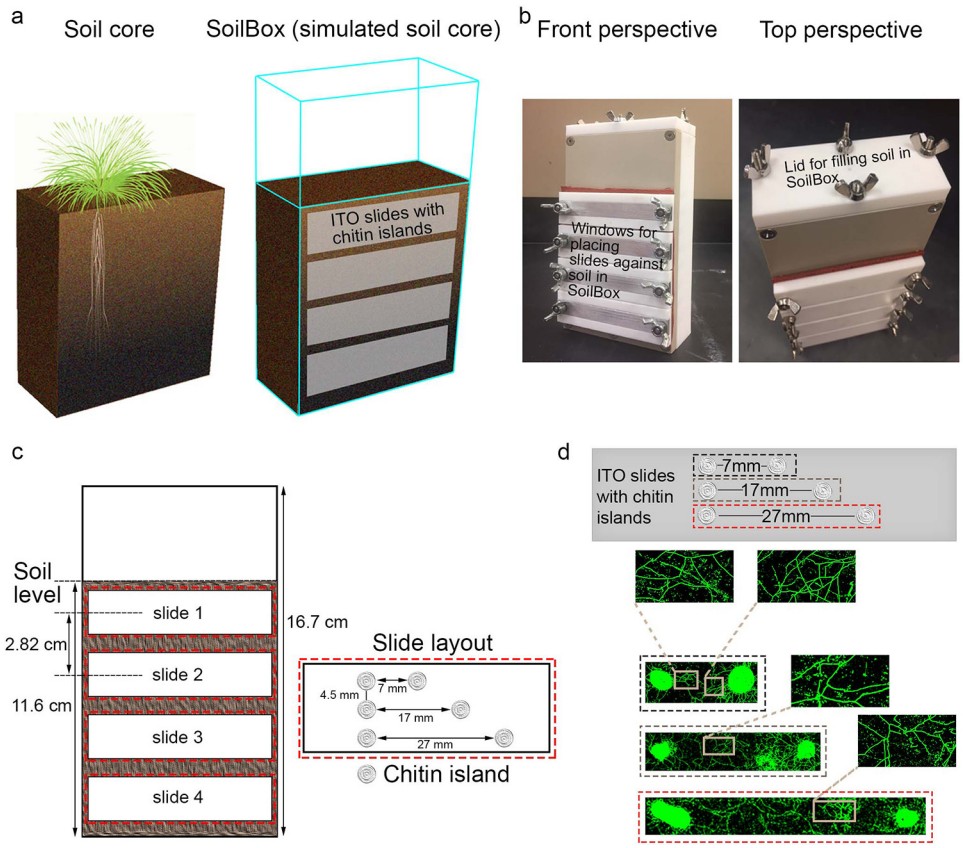

**FIG 1** Quantifying microbial community dynamics in a simulated soil ecosystem. (a) Scheme and concept of the SoilBox, inspired from a common soil core, where removable windows can be placed along the sagittal plane of the simulated soil core. Here, we employed functionalized indium tin oxide (ITO)-coated glass slides with printed chitin islands as our windows. (b) Image of a SoilBox used in this work, where we employed multiples of these boxes to test how microbial community structure and metabolism changed as a function of soil moisture regimes at different depths from the soil's surface. (c) Schematic of the SoilBox dimensions and slide slots. Soil is filled to a level of 11.6 cm. The center-to-center distances between the slides are 2.82 cm as shown in the figure. Schematic of a typical slide layout with chitin islands fabricated on an ITO-coated slide functionalized with aminopropyltri-methoxysilane (APTMS). Functionalizing of ITO-coated glass surfaces with APTMS rendered the ITO surface hydrophilic and amenable to colloidal chitin deposition by drop-casting. The center-to-center distances between the chitin islands laterally are 10, 20, and 30 mm (with 7.5-mm spacing of centers horizontally). The chitin islands were approximately 3 mm in diameter; therefore, the nearest points between the circumferences of two chitin islands laterally are 7, 17, and 27 mm. (d) Fungal mycelium bridging of nutrient sources over the different distances between chitin islands was observed only in the 14% moisture regime of slide 2, when stained with SYBR Gold (green). Data for the other moisture regimes can be seen in Fig. S2 and S3.

14%, 24%, and 34% soil moisture regimes (gravimetric moisture content). The 24% moisture regime was selected to represent the field capacity of the soil used in this study (23), whereas the 14% and 34% moisture regimes were introduced to probe the effects of drought and added moisture, respectively, on soil microbial processes. After incubating soil under the abovementioned moisture conditions for 7 days, ITO-coated glass slides fabricated with chitin islands were placed on the windows and incubated for 3 days, while keeping the respective moisture level consistent in each of the SoilBoxes. Under all conditions, the majority of the microbial biomass development was on the chitin islands compared to areas around the islands, as observed by confocal microscopy (Fig. S1 and S2). Interestingly, extensive fungal hyphal bridging was observed across the chitin islands (Fig. 1d) on slide 2 (Fig. 1b and c) under the drought-like condition of 14% moisture. These bridges were able to span distances between food sources of 27 mm within a 3-day time period. We did not observe such extensive fungal hyphal networks for the 24% and 34% soil moisture treatments (Fig. S2). These results support previous observations of fungal drought adaptability, where some fungi are

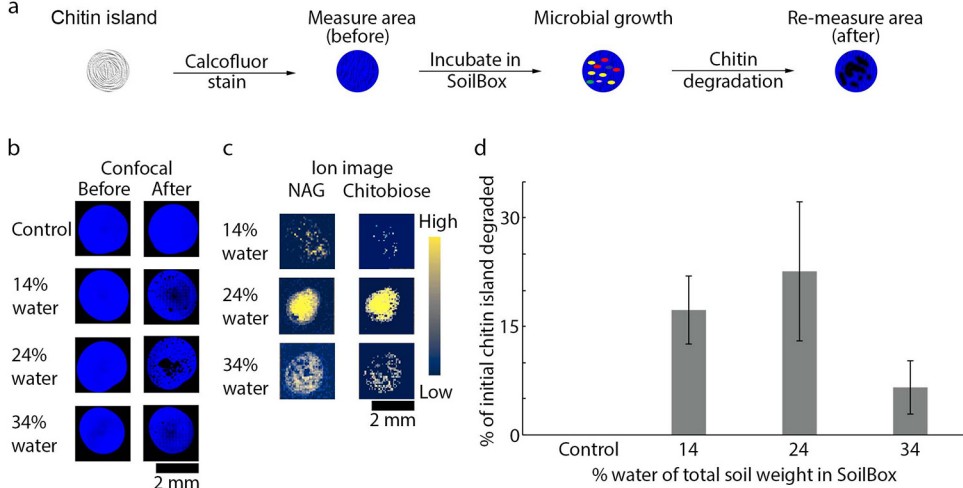

**FIG 2** Chitin degradation demonstrates growth and development of soil microbial communities. (a) A schematic demonstrating the process of quantifying the degradation of chitin on chitin islands as a result of microbial growth and development. (b) Degradation of chitin islands by the soil microbial communities as a result of 3-day incubation in the SoilBox was determined by measuring the area of degradation using confocal microscopy. Chitin islands, stained with calcofluor white (blue), were measured before and after incubation in the SoilBox to determine the amount of degradation (as annotated in the workflow). (c) The degradation products of chitin resulting from microbial growth on the chitin islands under different moisture regimes, NAG (*m/z* 222.0972) and chitobiose (*m/z* 425.1766), were detected using matrix-assisted laser desorption/ionization–mass spectrometry imaging (MALDI-MSI) in positive ion mode. (d) The quantitative analysis of the degradation of the calcofluor white-stained chitin islands (b) using confocal microscopy shows different amounts of degradation of chitin under different humidity regimes compared to the control, which was not exposed to microbial communities in the SoilBox. The error bars were generated from three different chitin islands from slide 2 per humidity condition (the standard deviation of these measurements). Note that the chitin islands used for confocal microscopy measurements (b and d) were different than the ones used for MALDI-MSI analysis (c).

capable of forming extensive networks of mycelial bridges across soil pores in dry and nutrient-poor habitats in order to access moisture and nutrients (24–26).

Using the SoilBox system also offered the advantage of investigating differences at different soil depths in microbial community structure and composition. Partial fungal hyphal networks were observed at the 14% moisture regime on slide 1 (center of slide, 1.57 cm from the soil surface) (Fig. S3), complete mycelial web formation was observed on slide 2 (center of slide, 4.39 cm from the soil surface) (Fig. 1d), and such networks were not observed on slide 4 (center of slide, 10.3 cm from the soil surface) (Fig. S3). These differences in the microbial biomass and morphology within the same simulated soil core are likely attributable to water drainage and presumable differences in oxygen concentrations toward the bottom of the SoilBox (27, 28). We did not observe such steep changes in community phenotype as a function of soil depth within the 24% and 34% moisture regimes. As such, our results demonstrate the ability of the SoilBox system to capture microbial community transformations induced by soil depth and microenvironmental changes within the same soil matrix. Worth highlighting is how the SoilBox system enabled, to our knowledge, the first spatial images of fungal mycelial food webs and community morphology changes occurring within a native soil microbial community.

Microbial growth under different moisture conditions induced degradation of the chitin islands over time, as quantified using confocal microscopy (Fig. 2a, b, and d). Significant changes in the relative quantities of chitin degradation due to microbial growth were observed over 3 days between drought and saturated soil moisture regimes (i.e., between 14% and 34%; $P < 0.05$). Chitin is a $\beta$-(1,4)-linked polymer of *N*-acetyl-D-glucosamine (NAG). The $\beta$-1,4-glycosidic bonds between the NAG residues that comprise a chitin chain are hydrolyzed by chitinases, a class of glycoside hydrolases produced by soil bacteria and fungi (29, 30). As such, we were able to track chitin decomposition via untargeted molecular imaging, where NAG and chitobiose were

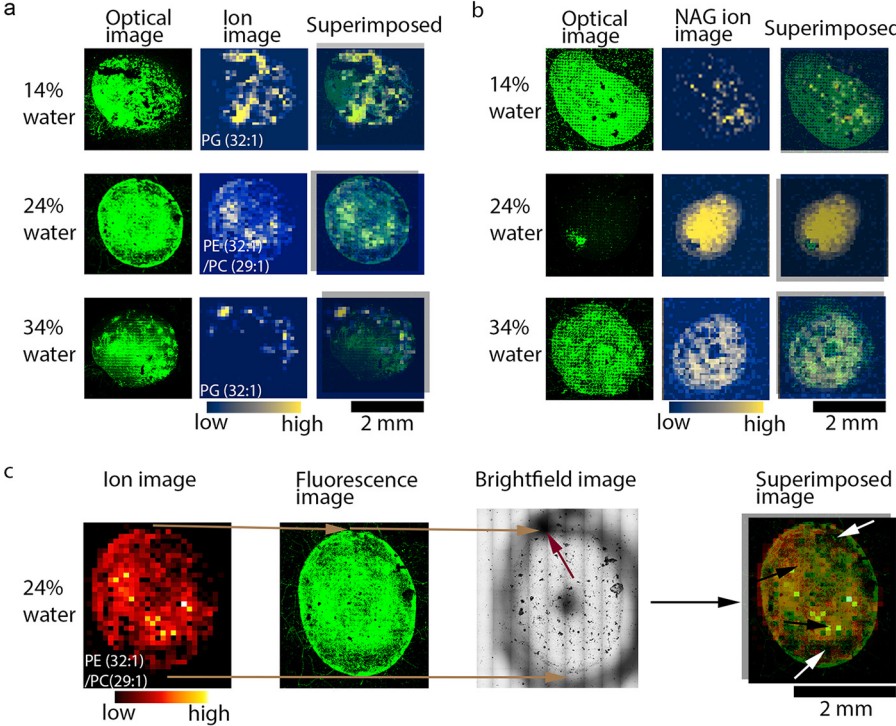

**FIG 3** Visualizing soil microbial processes with correlative image analysis. The spatial organization and metabolic snapshot of the soil microbiome were observed using confocal microscopy of SYBR Gold-stained (green) biomass and MALDI-MS molecular ion images, respectively, on the chitin spot. (a) The confocal microscopy images of biomass on chitin spots of slides incubated in SoilBoxes with 14%, 24%, and 34% (wt/wt) water to soil and the corresponding ion images of $m/z$ 719.4894, 688.4946, and 703.4945, which are putatively annotated as PG 32:1, PE 32:1, and PG 32:1, respectively. These were the most abundant lipids in their respective moisture regimes, so they were chosen as a representative ion image. (b) The confocal microscopy images of biomass on different chitin spots than those in panel a, from slides incubated in SoilBoxes with 14%, 24%, and 34% soil moisture, and the corresponding ion images of NAG ($m/z$ 222.0972). (c) Correlating MALDI-MS images and confocal microscopy images by overlaying the confocal microscopy biomass image on the corresponding MALDI-MS ion image, $m/z$ 688.4946 (putatively PE 32:1/PC 29:1), taken of the same spot on a chitin island on a slide incubated in a SoilBox with 24% soil moisture. The red arrow in the bright-field image (middle; grayscaled) and black circle around the chitin island (bright-field image) denote fiducial markers utilized for alignment of the ion and fluorescence images. The superimposed image (far right) shows areas of high (black arrows) and low (white arrows) biomass on the confocal image corresponding to higher- and lower-intensity lipid ion signals on the MALDI-MS image, respectively.

detected as products of microbial decomposition of chitin using MALDI-MSI (Fig. 2c). NAG and chitobiose were not detected on control chitin islands (i.e., islands not subjected to incubation in the SoilBox), suggesting that these degradation products resulted from biotic activity. We note that acid hydrolysis could also cause abiotic chitin degradation (31, 32), but the native soil in these SoilBoxes is alkaline (pH 8), which suggests that this effect would be minimal over the short time frame of the incubations.

Apart from imaging chitin degradation on chitin islands using the slide windows, the SoilBox was also designed to facilitate spatially explicit harvesting at different soil layers that correspond with the slide positions. This enabled a deep investigation of the biological and chemical interactions responsible for the spatial patterns revealed through imaging experiments. Probing of the incubated chitin islands using MALDI-MSI resulted in detection of several lipids (Fig. 3, Table S1, and Fig. S4), which can be used as biomarkers for microbial biomass growth and development within a soil ecosystem (33, 34). Several families of phospholipids, such as phosphatidylethanolamines (PE), phosphatidic acid (PA), phosphatidylglycerol (PG), phosphatidylcholine (PC), and phosphatidylinositol (PI), were observed on the chitin islands using MALDI-MSI under the different moisture regimes (Fig. 3a and c). Phospholipids are a major constituent of microbial cell membranes (35), and changes in environmental conditions can modulate

cell membrane phospholipid composition of microbial communities (36–38). As such, the most abundant lipids on the chitin islands were signatures of specific microorganisms that grew on chitin under the different moisture regimes (Fig. 3a and c). The complete list of phospholipid families and their relative abundances that were annotated under different moisture regimes are shown in Table S1, and representative average MALDI-MSI spectra of the chitin islands at 14%, 24%, and 34% moisture regimes are shown in Fig. S4. Most of the lipids putatively annotated (Table S1) were more prevalent at 24% moisture, with the exception of certain lipids like PI (32:0) and PI (18:4) that were more abundant in the 34% and 14% moisture regimes, respectively (Fig. S4b). Note that such lipids are not necessarily the most abundant lipids but rather are upregulated species observed under specific conditions. The most striking example is PE (38:5), which is completely absent from 14% and 34% moisture regimes but shows a strong MS signal under 24% moisture conditions (Fig. S4b). The detection of different phospholipids on the chitin islands incubated under the same conditions demonstrates the unprecedented spatial heterogeneity of soil microbial metabolism that was captured by using the SoilBox approach (Fig. S5) compared to other techniques.

To expand the usefulness of using multi-imaging modalities, we overlaid fluorescence images of microbial communities on chitin islands with ion images of lipid and NAG oligomers. This enabled us to create a spatial comparison between community biomass distribution and metabolic activity of the soil microbes (Fig. 3a and b). Bright-field images obtained simultaneously with fluorescence microscopy measurements provided easy visualization of fiducial markers that permit alignment of the microscopy and MS images (Fig. 3c). The ionic matrix composed of 2,5-dihydroxybenzoic acid and dimethylaniline (DHB-DMA) used for ionization of NAG oligomers provided the optimum ionization of NAG and NAG oligomers (Fig. 3b) (39). However, due to the low laser energy absorption coefficient of DHB, higher laser fluence for desorption ionization is required, and this can cause excessive sample damage during analysis (40). Accordingly, we used excessive laser fluence to cause significant ablation of the microbial biomass on the chitin, thus exposing the underlying NAG and chitobiose produced as a result of microbial activity. This resulted in higher-intensity signals of biomolecules and large areas of otherwise missing biomass as observed in the fluorescence microscopy experiments performed after MALDI-MSI analysis (Fig. 3b). The differences in the degree of microbial biomass attachment on the surface of the chitin islands under the various soil moisture conditions may explain why greater laser-induced damage was observed on the 24% chitin islands (Fig. 3b). Further investigation is required to determine if robustness of microbial biofilm formation on chitin under these conditions was the cause of inconsistent biomass removal.

The microbial biomass had little to no autofluorescence when imaged over larger areas, making pre-MALDI-MSI imaging of the SoilBox slides difficult (data not shown). Since the confocal microscopy images were obtained after MALDI-MSI analysis, it would be ideal to minimize sample damage from the laser ablation process to facilitate accurate comparison between areas of microbial biomass and corresponding ion images. By optimizing the laser power and number of shots required for desorption ionization, and by exploring different matrices, we were able to avoid these issues. We determined that sample preparation conditions using norharmane (Fig. 3a and c) and 1,5-diaminonaphthalene (DAN) (Fig. S6) yielded optimum signals in the case of ionization of lipids. It should be noted that we observed that the insulating properties of the chitin islands can also contribute to lipid ionization suppression (Fig. S7). However, as microbial communities develop on the chitin islands, such suppression would be presumably minimized due to degradation of chitin on the islands, as shown in Fig. 2.

Variations in microbial community dynamics within a soil ecosystem have previously been described using different omics studies (2), but such studies have yet to capture the variations imposed on microbial community dynamics at the microscale by a spatially diverse and changing soil landscape. In this work, we created a SoilBox system with field-relevant soil core dimensions, capable of simulating the environmental conditions of any soil ecosystem. This system provided direct visualization of the

organization of individual microbial communities and their metabolic activity. The use of multitechnique correlative imaging afforded spatial comparison of structural and metabolomic changes in soil microbial communities induced by environmental conditions (i.e., soil moisture). We anticipate that this platform will be invaluable in probing specific intra- and interkingdom soil microbial metabolic networks that can occur due to a variety of environmental stresses.

## MATERIALS AND METHODS

**SoilBox fabrication.** SoilBoxes were fabricated in-house, and more detailed schematics can be found in Fig. 1b and c and in the SoilBox blueprints in Fig. S8 in the supplemental material. Per the image of a SoilBox in Fig. 1b, the body was constructed using polytetrafluoroethylene (PTFE) Teflon (soft white material; Nationwide Plastics Inc.), the frame around the slide windows was fabricated from polyether ether ketone (PEEK) (hard tan material; McMaster-Carr), and the sealing gasket was created from 1/16-in.-thick silicone foam (McMaster-Carr). Stainless steel screws, wingnuts, and studs (McMaster-Carr) were used for the construction of the SoilBoxes. The slide windows were designed with dimensions to hold standard glass microscope slides (3 in. by 1 in. or 75 mm by 25 mm, depending on the manufacturer) against the soil surface. These slides can be removed at different time points to sample growth and visualize soil microbial processes.

**SoilBox preparation.** The SoilBox was designed to query soil microbial ecology under controlled conditions, with minimum disruption to native system dynamics at different depths from the soil's surface. The location of the slides represents soil depths just below the soil surface to a depth of 11.6 cm below the surface (slide window position 1 to 4, respectively [Fig. 1c]). Soil was filled to the level of 11.6 cm from the bottom of the SoilBox, providing a headspace of 5.5 cm above the soil level to the lid of the SoilBox, where the atmospheric conditions can be manipulated if needed. The SoilBoxes were positioned vertically as shown in Fig. 1b before filling in with soil through a lid that opens at the top surface of the box (Fig. 1b). The SoilBoxes were weighed before and after filling in with soil to measure the total weight of soil filled in each box.

Here, the SoilBoxes were filled with live soil collected in October 2017, from a field site operated by Washington State University, located in Prosser, WA, USA (46°15′04″N and 119°43′43″W) (23). This soil represents a Warden silt loam, characterized as a coarse-silty, mixed, superactive, and mesic Xeric Haplocambid. The soil is classified as a marginal soil, with low organic matter content (3.7%) and an average pH of ~8. At each site, bulk sampling was accomplished with a shovel within 0- to 20-cm depth from the ground, and samples were stored in plastic bags at 4°C. To exclude bigger soil aggregates and rocks, samples were sieved with 4-mm mesh size.

A portion of the soil collected from the field was weighed, dried overnight at 65°C, and reweighed. The change in weight was measured to gauge the water content in the field soil—determined to be 4% by weight. Dry soil was used to fill the SoilBoxes, and water was added to soil using a syringe automation device (intranasal automation drug delivery; Teleflex), which created a gentle plume that did not disturb the top of the soil bed, to generate environments with 14%, 24%, and 34% moisture (based on percent total weight of the soil used to fill the SoilBox). The SoilBoxes were incubated at 25°C in an Innova 42 incubator (New Brunswick Scientific) for 7 days for conditioning of microbial biomass to different moisture regimes, following which the ITO slides (see preparation details below) were placed in the SoilBox windows (Fig. 1b) fabricated for placing slides against the soil surface.

**Fabrication and utilization of chitin-functionalized ITO-coated slides.** Colloidal chitin was prepared using a previous protocol with slight modifications (22). Briefly, a 0.4% (wt/vol) colloidal chitin solution was made by mixing beta chitin powder (Synquest Laboratories) in hexafluoroisopropanol (Sigma-Aldrich) and stirred for 5 days under ambient conditions until all the powdered chitin went into solution. ITO-coated glass slides (Bruker Daltonics), which are electrically conductive but retain high optical transparency, were cleaned using 5-min subsequent acetone-methanol (MeOH)-isopropyl alcohol (IPA)-water ultrasonication steps to remove any surface contamination and then were plasma cleaned for 3 min to remove all organic impurities. The slides were functionalized by immersion in 4% (wt/vol) aminopropyltrimethoxysilane in ethyl alcohol (EtOH) for 15 h and then hard baked at 180°C for 30 min, in order to make the ITO surface hydrophilic for chitin deposition. The chitin island pattern (schematic in Fig. 1c) was created in AutoCAD and then laser cut into a Teflon membrane. This island template membrane was placed on functionalized ITO-coated slides, and 5 $\mu$l of colloidal chitin was drop-cast into each of the cutout positions. After this, the slides were placed in a covered petri dish and slowly dried. The slow evaporation ensured attachment of chitin to the functionalized ITO-coated slides.

Finally, the slides were placed in the windows provided for slides in the prepared SoilBoxes (see above), where they were incubated for 3 days before removal for analysis. During the incubation period, the moisture level was kept consistent by maintaining a constant weight of the SoilBoxes that matched their respective initial weights. Once the slides were detached from the SoilBox, larger soil particles were removed from the slides using compressed nitrogen. Finally, the slides were dried in a vacuum desiccator overnight before MALDI-MSI sample preparation and analysis.

**MALDI-MSI sample preparation, data acquisition, and analysis.** Matrix application of the 3-day-incubated ITO-coated slides was performed using a TM-sprayer (HTX Technologies) (41). The optimum cycles of matrix application, matrix solvent, and spray temperature were determined by trial and error for ionization of different ion species. For measuring lipid content in negative ion mode, 7 mg/ml of norharmane (Sigma-Aldrich) dissolved in 2:1 $CHCl_3$-MeOH was sprayed with 9 passes at a flow rate of 120 $\mu$l/min, at 30°C, and with a spray spacing of 3 mm and a spray velocity of 1,200 mm/min. For imaging

chitin degradation in positive ion mode, an ionic matrix was created using 2,5-dihydroxybenzoic acid (DHB; Sigma-Aldrich) and dimethylaniline (DMA; Sigma-Aldrich) as described before (39). Briefly, matrix solution was prepared as an equimolar mixture of DHB and DMA (100 mg/ml DHB in $H_2O$-acetonitrile [ACN]-DMA, 1:1:0.02) and sprayed with 4 passes at a flow rate of 50 $\mu$l/min, at 80°C, and with a 3-mm spray spacing and a spray velocity of 1,200 mm/min. Additional MALDI sample preparation details can be found in the supplemental methods (Text S1).

MALDI-MSI data were acquired using a 15 Tesla Fourier transform ion cyclotron resonance mass spectrometer (FTICR-MS; Bruker Daltonics) equipped with a Smartbeam II laser (355 nm, 2 kHz) using 150 laser shots/pixel and 100-$\mu$m pitch between pixels. The MS was externally calibrated using tune-mix in electrospray ionization (ESI) mode in an effort to provide mass accuracy of ~1 ppm or less. For lipid analysis, the FTICR-MS was operated to collect ions with *m/z* 400 to 1,800 in negative ion mode, using an 0.9-s transient, which resulted in a mass resolution (*R*) of ~260,000 at 400 *m/z*. For chitin oligomer analysis, the FTICR-MS was operated to collect *m/z* 200 to 2,000 in positive ion mode using an 0.5-s transient, which resulted in an *R* of ~140,000 at 400 *m/z*. Data were acquired using FlexImaging (v. 4.1; Bruker Daltonics), and image processing and visualization were performed using SCiLS Lab (Bruker Daltonics). The molecules were putatively annotated using the METLIN metabolite database using a mass error of 3 ppm (42).

**Confocal microscopy sample preparation, data acquisition, and analysis.** Calcofluor white dye (Sigma-Aldrich) was used for visualizing chitin degradation, where 100 $\mu$l (1 g/liter) of the calcofluor white solution was added to the chitin islands prior to incubation in the SoilBox. Slides were washed with deionized (DI) water to removed excess stain. The chitin islands were imaged prior to slide incubation in the SoilBoxes containing the different moisture regimes and then again after 3 days in the SoilBoxes. A control slide, with stained chitin islands that were never incubated in a SoilBox, was imaged in conjunction, before and after the incubation period, to account for any instrumental drift between analyses. For visualizing microbial biomass growth on the chitin islands, the ITO-coated slides were washed in EtOH for 2 min to remove the matrix, after MALDI-MSI analysis, and then were stained with 100 $\mu$l of a 1:5,000 dilution of SYBR Gold dye (Sigma-Aldrich) for 15 min and rinsed with DI water before confocal microscopy analysis.

All confocal microscopy images were acquired using a Zeiss 710 microscope and a W plain-Apochromat 20× objective. Calcofluor white and SYBR Gold dye were excited with 405- and 490-nm laser wavelengths, respectively, and the pinhole was adjusted to 4 Airy units. Emission wavelengths were collected from 410 to 523 nm and 499 to 635 nm for the respective analyses. Using the tile scan function in ZEN 2.3 SP1 software (Zeiss), several 400-$\mu$m by 400-$\mu$m images were stitched over different areas on the slides to acquire mosaic images of chitin islands and several areas around and between the islands. All confocal microscopy images were analyzed using the ZEN image analysis software (Zeiss). The calcofluor white-stained images of chitin islands were analyzed using the Volocity 6.3 image analysis software (PerkinElmer), where the pixel area before and after incubation was calculated for three different chitin islands per humidity condition (14%, 24%, and 34% of total soil weight, respectively). The error bars in Fig. 2d represent the standard deviation. The differences between the before and after images were used to calculate the amount of chitin degraded by the soil microbial community. The difference in pixel area of the control sample chitin area, before and after, was subtracted from the final amount of chitin degraded, in an effort to normalize differences in laser power of the confocal microscope.

The MSI and confocal images were compared using a procedure reported previously (43). Briefly, for aligning the ion images with the fluorescence intensity images of a chitin island, bright-field images were obtained simultaneously (Fig. 3c shows an example). The outline around the chitin islands and the top of the chitin islands were marked with a circle (black circle) and an arrow (shown with red arrow in Fig. 3c), respectively, which served as fiducial markers to align the fluorescence and ion images. These fiducials were also used to register the mass spectrometer to the sampling area, where analysis areas were chosen for MALDI-MSI. Using these fiducial markers, overlays for comparison of MSI and confocal images were constructed.

## SUPPLEMENTAL MATERIAL

Supplemental material is available online only.

**TEXT S1**, DOCX file, 0.02 MB.
**FIG S1**, JPG file, 0.3 MB.
**FIG S2**, JPG file, 1 MB.
**FIG S3**, JPG file, 1.8 MB.
**FIG S4**, JPG file, 0.9 MB.
**FIG S5**, JPG file, 0.7 MB.
**FIG S6**, JPG file, 1.7 MB.
**FIG S7**, JPG file, 1.2 MB.
**FIG S8**, PDF file, 0.3 MB.
**TABLE S1**, DOCX file, 0.02 MB.

## ACKNOWLEDGMENTS

This research was supported by the Department of Energy (DOE) Office of Biological and Environmental Research (BER) and is a contribution of the Scientific Focus Area "Phenotypic response of the soil microbiome to environmental perturbations." Pacific Northwest National Laboratory (PNNL) is operated for the DOE by Battelle Memorial Institute under contract DE-AC05-76RLO1830. A portion of the research was performed using the Environmental Molecular Sciences Laboratory, a DOE Office of Science User Facility sponsored by the BER and located at PNNL.

A.B., D.V., K.S.H., and C.R.A. designed the experiments. A.B., D.V., and S.L.B. performed the experiments. C.R.A. and T.W.W. designed the SoilBox, and T.W.W. constructed the SoilBox. C.R.A., K.S.H., and J.K.J. supervised all the experiments. A.B. and C.R.A. wrote the manuscript. All authors contributed to editing the manuscript.

The authors declare no competing interests.

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
