## [Reviewer comments · mSystems]

Visualizing microbial community dynamics via a controllable soil environment

Arunima Bhattacharjee, Dusan Velickovic, Thomas Wietzma, Sheryl Bell, Janet Jansson, Kirsten Hofmockel, and Christopher Anderton

Corresponding Author(s): Christopher Anderton, Pacific Northwest National Laboratory

Review Timeline:

Submission Date:	October 3, 2019
Editorial Decision:	December 17, 2019
Revision Received:	January 21, 2020
Accepted:	January 23, 2020

Editor: Paul Wilmes

Reviewer(s): The reviewers have opted to remain anonymous.

Transaction Report:

DOI: <https://doi.org/10.1128/mSystems.00645-19>

December 17, 2019

Dr. Christopher R Anderton
Pacific Northwest National Laboratory
Richland 99352

Re: mSystems00645-19 (Visualizing microbial community dynamics via a controllable soil environment)

Dear Dr. Christopher R Anderton:

Please revise your manuscript in line with the comments by reviewer 2.

Below you will find the comments of the reviewers.

To submit your modified manuscript, log onto the eJP submission site at <https://msystems.msubmit.net/cgi-bin/main.plex>. If you cannot remember your password, click the "Can't remember your password?" link and follow the instructions on the screen. Go to Author Tasks and click the appropriate manuscript title to begin the resubmission process. The information that you entered when you first submitted the paper will be displayed. Please update the information as necessary. Provide (1) point-by-point responses to the issues raised by the reviewers as file type "Response to Reviewers," not in your cover letter, and (2) a PDF file that indicates the changes from the original submission (by highlighting or underlining the changes) as file type "Marked Up Manuscript - For Review Only."

Please return the manuscript within 60 days; if you cannot complete the modification within this time period, please contact me. If you do not wish to modify the manuscript and prefer to submit it to another journal, please notify me of your decision immediately so that the manuscript may be formally withdrawn from consideration by mSystems.

To avoid unnecessary delay in publication should your modified manuscript be accepted, it is important that all elements you upload meet the technical requirements for production. I strongly recommend that you check your digital images using the Rapid Inspector tool at <http://rapidinspector.cadmus.com/RapidInspector/zmw/>.

Sincerely,

Paul Wilmes

Editor, mSystems

Journals Department
Reviewer comments:

Reviewer #2 (Comments for the Author):

Bhattacharjee et al. custom-built and utilized the SoilBox system to study the spatial aspects of the soil microbial activity and interactions in different soil moisture conditions, as a demonstration of the SoilBox's potential to enable the multi-modal and -omics interrogation of the soil microbial ecosystem. The reviewer finds the SoilBox system a clever and useful device for such usage, especially made more valuable by its access to the molecular imaging capability (MALDI-MSI). In this case study with chitin as a nutrient substrate, the authors observed the moisture and soil-depth dependence on fungal mycelium network formation as well as the moisture dependence on microbial activity. Also, the various phospholipids were observed using MALDI-MSI, and spatial heterogeneity in microbial activity. Some comments:

1. Figures 1 and S1 can be combined for readability. The manuscript is about introducing the SoilBox system and having a more detailed description as in S1 and the real picture of the device in the main manuscript would be helpful. Also, the notion of slide #1, 2, 3, and 4 was difficult to understand just from the Figure 1, something that I didn't want to necessarily refer to the supporting information to get the idea.
2. Figure 2 is very hard to understand, as it combines the graph and pictures together in a disorganized way. This needs to be re-configured for better readability.
3. Figure 2: What is the parameter for the error bar? Size of n?
4. Figure 3B: Why is the confocal image of 24% moisture sample so dim compared to other spots? Aren't they all similarly damaged by the laser?
5. Figure 1B in comparison to Figure S2, S3, and S4. The fluorescence intensity is much brighter in Figure 1B compared to any other figures. Are the fluorescence intensity normalized across the manuscript, or indicated otherwise?
6. Figure S2 and S3, the 34% moisture soil seems to have more mycelium network compared to 24% moisture soil. Any comments on this trend? Also, in Figure S3, are these images from Slide #2 as well, like that from 14% moisture soil?
7. Showing some representative MALDI spectra from the chitin spot in the manuscript (or SI), not just from the model system in Figure S7.
8. Readability issue: information like 7-day incubation + 3-day incubation can be explained earlier in the Result/Discussion section, not just in the Method section for readability.

Microbial cell products/necromass are major constituents of soil carbon. Various environmental/edaphic factors influence microbial processes, which can have an impact on the global carbon cycle. In situ field study of microbial response to changing environmental conditions is challenging. Similarly, incubation experiments done in the lab/greenhouse fail to capture the broad variations (spatial/temporal) within the system. The technique used in this study is promising and could be implemented to study microbial response to the changing environment at the temporal and spatial scale. However, there are concerns and limitations of the experimental approach and the technique used in the study. I would recommend authors to have a separate paragraph in the main text that discusses the challenges and limitations of the study so that the readers can have a clear idea. I would recommend accepting the manuscript after major changes.

Comment 1:

I feel like the method section can be simplified and improved. I found it confusing when reading about how the soil was prepared before filling soilboxes. Was the soil homogenized after collection? I assume storing and sieving processes must have mixed the soil within the collected depth. [Line 223 – 225: “The location of the slides represents soil depths just below the soil surface to a depth of 11.6 cm below the surface (slide window position 1 to 4, respectively, FIG. S1b).”

Line 235-238: “At each site, bulk sampling was accomplished with a shovel within 0 to 20 cm depth from the ground and it was stored in plastic bags at 4 °C. To exclude bigger soil aggregates and rocks, samples were sieved with 4 mm mesh size.”]

Similarly, do you mean soil dried at 65 °C was used to fill the soilboxes? [Line 241: “Dry soil was used to fill the SoilBoxes”]. I hope that was not the case. I don’t think treating soil at 65C would preserve the microbial community as present in the field.

Comment 2:

More discussion is needed on why extensive hyphal growth was only seen on slide 2 and not others? Is it because when watering from the top there is significant water gradient within soilboxes? It would be nice to see moisture content data from soil adjacent to the slides, especially for drought system as there can be a significant difference in water content/potential. I assume that for 14% system there is very little water (water<<14%) to support good microbial growth below slide 2 and slide 1 most likely not experiencing true drought (water>>14%). This problem in some degree beats the purpose of having three different systems as there possibly is an extensive water gradient within each system. It would have been better if water content was adjusted and the soil was mixed before the boxes were filled. Under the current set up, the true moisture content in the soil adjacent to slid would be very helpful. This will put things in a better perspective for the readers.

Comment 3:

I am skeptical that the method/technique used in this study can identify many different phospholipids at a resolution where it can be used to short out diverse microbial community dynamics as mentioned in lines 152-167.

Line 42: better to list 11.6 rather than ~12.

Line 72: To address these challenges...., let's be honest, it would be better to say "to address some of these challenges" as the study does not have good control for all the challenges listed in the previous paragraph.

Line 96- I would say "depth profile most soil ecologists".

Line 99: Please mention the distance (in parentheses). This would better serve the readers rather than having to find a figure in supporting information.

Line 100: Define "ITO"

Line 102: Maybe when mentioning why chitin was chosen, also mention that it is different from the bacterial products.

Line 103: How about a line mentioning the limitation of using Chitin, "however we are aware that the use of a specific OC source can have some influence on substrate-based selection on the microbial community" something along this line would be good.

Line 105: "Regulates" might be to a strong statement, maybe use "influences".

Line 108: What was the saturation (100% water holding capacity) water content? Please list that somewhere. It will be helpful for readers to compare with studies that mention water content in water holding capacity. I am assuming 34% gravimetric is probably very close to 100% water holding capacity.

Line 112: Combine S2 and S3 with Figure 1B to the main text (if possible). If you want to keep as it is then referencing Figure 1B along with S2 and S3.

Figure 1S: What was on slide 4? Was it a blank slide? Not clear from the figure or the caption.

Line 127 – 129: "We do not observe such steep changes in community phenotype as a function of soil depth within the 24% 128 and 34% moisture regimes." Please mention the possibility of not having a significant water gradient in these systems unlike in 14%. Again, moisture content for soil next to the slide would be great.

Line 270: What is TM?

Figure 3A: Please mention the slide number next to the images.

Little more discussion about other systems except the 14% would be nice too. I understand it was less interesting, but some more discussion would have been better.

Responses to Reviewers' Comments to Author

We thank the Editor and the Reviewers for taking time to diligently look over this manuscript, and provide comments that, we feel, made it a much stronger manuscript. Along with this document, we have resubmitted two versions of the manuscript, one with the changes annotated in blue font and a clean version with those changes incorporated, to ease the rereview of this manuscript.

Reviewer #2 (Comments for the Author):

Bhattacharjee et al. custom-built and utilized the SoilBox system to study the spatial aspects of the soil microbial activity and interactions in different soil moisture conditions, as a demonstration of the SoilBox's potential to enable the multi-modal and -omics interrogation of the soil microbial ecosystem. The reviewer finds the SoilBox system a clever and useful device for such usage, especially made more valuable by its access to the molecular imaging capability (MALDI-MSI). In this case study with chitin as a nutrient substrate, the authors observed the moisture and soil-depth dependence on fungal mycelium network formation as well as the moisture dependence on microbial activity. Also, the various phospholipids were observed using MALDI-MSI, and spatial heterogeneity in microbial activity. Some comments:

1. Figures 1 and S1 can be combined for readability. The manuscript is about introducing the SoilBox system and having a more detailed description as in S1 and the real picture of the device in the main manuscript would be helpful. Also, the notion of slide #1, 2, 3, and 4 was difficult to understand just from the Figure 1, something that I didn't want to necessarily refer to the supporting information to get the idea.

Response: We thank the Reviewer for this constructive comment. We have followed their advice very closely and combined the figures they are refereeing to. Now, FIG. 1 and FIG. S1 are reformatted into FIG. 1.

2. Figure 2 is very hard to understand, as it combines the graph and pictures together in a disorganized way. This needs to be re-configured for better readability.

Response: Upon further examining this figure, we concur with the Reviewer– it was confusing. We reformatted this figure for better readability.

3. Figure 2: What is the parameter for the error bar? Size of n?

Response: The error bar in the current (reformatted) FIG. 2d is from the slide 2 of three different chitin islands per humidity condition. We have included a detailed explanation for this in the methods section '**Confocal microscopy sample preparation, data acquisition and analysis**'.

4. Figure 3B: Why is the confocal image of 24% moisture sample so dim compared to other spots? Aren't they all similarly damaged by the laser?

Response: All the samples analyzed using DHB-DMA matrix were similarly damaged by the laser. The difference in the biomass damage observed in different samples in FIG. 3b, we attribute to a function of how well attached the microbial biomass was to the chitin islands, where loosely attached microbial biomass is easily removed by laser induced damage. The attachment of microbial biomass or robustness of biofilm formation on chitin islands is, perhaps a function of different humidity in the soil matrix and requires further investigation– something beyond the scope of this manuscript. We included more explanation regarding this in the results and discussion section of the manuscript, so that it is clear to the reader.

5. Figure 1B in comparison to Figure S2, S3, and S4. The fluorescence intensity is much brighter in Figure 1B compared to any other figures. Are the fluorescence intensity normalized across the manuscript, or indicated otherwise?

Response: The brightness contrast of the FIG. 1b was altered using a thresholding function to make the mycelia bridging across the chitin island more visible. We thank the Reviewer for this constructive comment and have applied the same thresholding function to the rest of the fluorescence images in the manuscript to make them comparable.

6. Figure S2 and S3, the 34% moisture soil seems to have more mycelium network compared to 24% moisture soil. Any comments on this trend? Also, in Figure S3, are these images from Slide #2 as well, like that from 14% moisture soil? From slide 2

Response: We thank the Reviewer for pointing out this interesting trend in mycelium network growth. We observed this trend of increased mycelium growth in slide # 2 of 14% and 34% humidity conditions in several biological replicates, and we believe this to be a function of microbial community dynamics under certain humidity regime. However, without the knowledge of the community composition in these experimental conditions, it is difficult to conclude anything further regarding this trend. The 14% soil moisture is a drought-like condition and fungal communities show increased bridging to access nutrients and water under low humidity. We also observed increase in fungal growth around the chitin islands and incompletely bridging at 34% moisture. We cannot comment on the increase in fungal growth at 34% soil moisture without further investigating the community composition at this experimental condition, which was beyond the scope of this manuscript.

The images in FIG. S3 are also from slide 2. We have included this information in the figure caption to clarify this confusion.

7. Showing some representative MALDI spectra from the chitin spot in the manuscript (or SI), not just from the model system in Figure S7.

Response: We have now included average MALDI spectra of chitin islands at 14%, 24%, and 34% soil moisture analyzed using norharmane matrix as FIG. S4. In table S1, we changed the

last column to reflect the relative abundance ratios of the lipids under different humidity condition.

8. Readability issue: information like 7-day incubation + 3-day incubation can be explained earlier in the Result/Discussion section, not just in the Method section for readability.

Response: We thank the reviewer for this comment and have included the incubation details in the results and discussion section.

January 23, 2020

Dr. Christopher R Anderton
Pacific Northwest National Laboratory
Richland 99352

Re: mSystems00645-19R1 (Visualizing microbial community dynamics via a controllable soil environment)

Dear Dr. Christopher R Anderton:

Your manuscript has been accepted, and I am forwarding it to the ASM Journals Department for publication. For your reference, ASM Journals' address is given below. Before it can be scheduled for publication, your manuscript will be checked by the mSystems senior production editor, Ellie Ghatineh, to make sure that all elements meet the technical requirements for publication. She will contact you if anything needs to be revised before copyediting and production can begin. Otherwise, you will be notified when your proofs are ready to be viewed.

Sincerely,

Paul Wilmes
Editor, mSystems

Journals Department
FIG S7: Accept
FIG S3: Accept
Supplemental Methods: Accept
SoilBox_blueprints_1: Accept
FIG S1: Accept
Table S1: Accept
FIG S6: Accept
FIG S4: Accept
FIG S2: Accept
FIG S5: Accept